

# Association behavior between sand tiger sharks and round scad is driven by mesopredators

Nicholas C. Coleman[1,2,*] and Erin J. Burge[1,*]

[1] Department of Marine Science, Coastal Carolina University, Conway, SC, United States of America
[2] Chesapeake Biological Laboratory, University of Maryland Center for Environmental Sciences, Solomons, MD, United States of America
[*] These authors contributed equally to this work.

## ABSTRACT

In marine systems, behaviorally-mediated indirect interactions between prey, meso-predators, and higher trophic-level, large predators are less commonly investigated than other ecologic interactions, likely because of inherent difficulties associated with making observations. Underwater videos ($n = 216$) from SharkCam, a camera installation sited beneath Frying Pan Tower, a decommissioned light house and platform, on a natural, hard bottom site approximately 50 km off Cape Fear, North Carolina, were used to investigate association behavior of round scad *Decapterus punctatus* around sand tiger sharks *Carcharias taurus*. Videos containing sand tiger sharks were analyzed for the simultaneous presence of round scad, and six species of scad mesopredators, with scad-shark interactions assigned to one of three categories of association: no visible interaction, loosely associated, or tightly associated. The likelihood of scad being loosely or tightly associated with sharks was significantly higher in the presence of scad mesopredators, suggesting that sharks provide a predation refuge for scad. This behaviorally-mediated indirect interaction has important implications for trophic energy transfer and mesopredator control on hard bottoms, as scad are one of the most abundant planktivorous fish on hard bottoms in the western Atlantic Ocean. Although we were not able to provide statistical evidence that sand tiger sharks also benefit from this association behavior, we have clear video evidence that round scad association conceals and attracts mesopredators, enhancing predation opportunities for sand tiger sharks. These interactions potentially yield additional trophic consequences to this unique association and highlight the value of exploring behaviorally-mediated interactions in marine communities.

# INTRODUCTION

Multispecies interactions highlight the underlying interdependency between organisms that exist in all communities (*Hutchinson, 1959*; *Paine, 1984*; *Beard & Dess, 1988*). These interactions are often categorized as direct, between two species, or indirect, where the interaction between two species ultimately affects a third species (*Wootton, 1993*).

Corresponding author
Erin J. Burge, eburge@coastal.edu

Behaviorally-mediated indirect interactions (BMIIs) are a further classification of trait-mediated indirect interactions (TMIIs) that are regulated by changes in a species' behavior (*Dill, Heithaus & Walters, 2003*). Both direct and indirect interactions produce changes in a species' density or behavior and are useful in understanding food webs and trophic exchanges. Indeed, indirect affects alter the risk landscapes of prey, including mesopredators which are prey species for larger carnivores (*Ritchie & Johnson, 2009*), especially in aquatic ecosystems (*Preisser, Bolnick & Benard, 2005*; *Heupel et al., 2014*), and can initiate trophic cascades (*Schmitz, Krivan & Ovadia, 2004*; *Creel & Christianson, 2008*). Competitor facilitation has been used in the context of BMIIs when the presence of one species of predator causes a change in the behavior of a prey species that makes that prey species more accessible for a second species of predator (*Dill, Heithaus & Walters, 2003*). For example, some marine, demersal mesopredatory fishes access pelagic prey that are driven towards the seafloor by pelagic mesopredators, enhancing feeding opportunities (*Auster et al., 2009*; *Auster et al., 2013*; *Campanella et al., 2019*). Despite their importance, BMIIs are less commonly investigated than other ecologic interactions because of the difficulty attributed to quantifying changes in behavior, especially within marine habitats.

## Underwater observations

Underwater videography using stationary cameras is an efficacious method to conduct in-situ marine observations, including surveying marine fish assemblages for species richness and abundance, and behavioral observations (reviewed in *Mallet & Pelletier, 2014*). For example, video has revealed that hard bottom habitat near Frying Pan Tower, 50 km offshore of Cape Fear, North Carolina, supports a diverse assemblage of both temperate and tropical reef fish species that fluctuates seasonally (*Burge et al., 2012*; *Burge & O'Brien, 2020*). With few exceptions, the use of underwater video to assess multispecies behavior that includes large and highly mobile predators, is much more limited (but see *Davis et al., 1999*; *Dunbrack & Zielinski, 2003*; *Barker, Peddemors & Williamson, 2011*; *Bond et al., 2012*; *Kanno et al., 2019*; *Brown et al., 2020*), despite evidence that video may be an advantageous survey method (*McCauley et al., 2012*).

Once recorded, videos can be reviewed multiple times to optimize the amount of data collected from a single event, and can potentially increase recognition of behaviors that would otherwise be difficult to assess during SCUBA diving observations, which are depth and time limited, especially in multispecies interactions where participants may have different reactions to the presence of observers. Potential bias associated with SCUBA surveys can be introduced as fish react to the presence of divers, sometimes for long periods post-survey (*Emslie et al., 2018*), and this effect has been observed to be species specific (*Cole, 1994*; *Kulbicki, 1998*; *Burge et al., 2012*; *Lindfield et al., 2014*). For example, sand tiger sharks display increased respiration and movement in the presence of SCUBA divers (*Barker, Peddemors & Williamson, 2011*), providing evidence that underwater stationary videography may be a better alternative to observe the behavior of this species.

## Sand tiger sharks (STs)

Sand tiger sharks, *Carcharias taurus* Rafinesque, 1810, are large (to >4 m), heavy-bodied lamniforms found in coastal and continental shelf waters of warm-temperate and tropical

seas worldwide. The diet of *C. taurus* is diverse and dominated by bony fishes and other elasmobranchs (*Gelsleichter, Musick & Nichols, 1999*). Individuals and aggregations occur especially in coastal areas of Australia (where they are known locally as grey nurse sharks), the east coast of South Africa (raggedtooth sharks), and the east coast of the United States (sand tiger sharks). Sand tiger shark aggregations are associated with migratory behavior, feeding, and reproduction (*Compagno, 2001*), and complex social networks and behaviors are described in this species (*Haulsee et al., 2016a*). Year-to-year site fidelity on shipwrecks and other artificial structures has been reported in North Carolina (*Paxton et al., 2019*) and individuals tagged in Delaware Bay undergo long-distance migrations to overwinter in continental shelf waters of North Carolina (*Teter et al., 2015*) and return to Delaware Bay predictably (*Haulsee et al., 2016b*). Additionally, sand tiger sharks are one of the few large-bodied sharks commonly housed in captivity (*Govender, Kistnasamy & Van Der Elst, 1991*; *Gordon, 1993*; *Smale et al., 2012*).

The conservation status of *C. taurus* is listed as globally Vulnerable (*Pollard & Smith, 2009*) with some regional populations considered endangered or critically endangered (*Cavanagh et al., 2003*; *Chiaramonte, Domingo & Soto, 2007*). Assessment of *C. taurus* in the northwestern Atlantic has suggested this population declined precipitously since the 1970s from overfishing in the 1980s to the mid-1990s (*Musick, Branstetter & Colvocoresses, 1993*; *Musick et al., 2000*), but analyses of multiple datasets now suggest only low to moderate declines in abundance (0.2–6.2%) (*Carlson et al., 2009*), in conflict with previous reports. More recent data suggest that the northwestern Atlantic population may be stabilized (*Frazier, Paramore & Rootes-Murdy, 2019*; *Latour & Gartland, 2020*). Regardless, conservative management is recommended due to the very low productivity for this species (*Goldman, Branstetter & Musick, 2006*; *Carlson et al., 2009*). Harvest of sand tigers is currently prohibited in the United States under Highly Migratory Species regulations and Habitat Area of Particular Concern designations are under further consideration (*NOAA Fisheries, 2019*).

## Round scad (RS)

Round scad, *Decapterus punctatus* (Cuvier, 1829), are small-bodied (<300 mm) carangids found in the western Atlantic Ocean southward from Nova Scotia, at Bermuda, and in the Gulf of Mexico, Caribbean Sea, and along continental shores of South America to Rio de Janiero (*Naughton, Saloman & Vaught, 1986*). In the South Atlantic Bight (SAB), round scad (typically 60–170 mm) are abundant in continental shelf waters in summer and fall, and move to hard bottoms in deeper, warmer areas of the mid- and outer shelf in winter (*Hales Jr, 1987*). Throughout their range, adult round scad are diurnal zooplankton specialists with the diet dominated by pelagic species and life stages (for meroplankton), as opposed to demersal plankton (*Hales Jr, 1987*; *Donaldson & Clavijo, 1994*).

Round scad are very frequently reported as stomach contents of pelagic and demersal piscivores (*Matheson, Huntsman & Manooch, 1986*; *Naughton, Saloman & Vaught, 1986*), and are often the most abundant species on FADs (*Rountree, 1990*), wrecks (*Lindquist & Pietrafesa, 1989*), and natural live bottoms (*Parker, Chester & Nelson, 1994*; *Kendall, Bauer & Jeffrey, 2009*; *Burge et al., 2012*) in portions of the SAB. Round scad schooling behaviors

around artificial structures (*Rountree, 1989*; *Rountree, 1990*; *Lindquist & Pietrafesa, 1989*; *Rountree & Sedberry, 1991*), while feeding (*Rountree & Sedberry, 1991*), and in response to their predators (*Auster et al., 2009*; *Auster et al., 2013*; *Campanella et al., 2019*) suggest an affinity for physical objects which can include larger fishes and elasmobranchs (*Fuller & Parsons, 2019*).

## Mesopredators (MPs)

A diverse assemblage of pelagic and demersal piscivorous mesopredatory fishes inhabit hard bottom habitats of the SAB (*Chester et al., 1984*; *Sedberry & Van Dolah, 1984*; *Kendall, Bauer & Jeffrey, 2009*; *Burge et al., 2012*; *Burge & O'Brien, 2020*). Pelagic mesopredators include medium to large-sized jacks (e.g.: greater amberjack *Seriola dumerili* (Risso, 1810), almaco jack *Seriola rivoliana* Valenciennes in Cuvier and Valenciennes, 1833, blue runner *Caranx crysos* (Mitchill, 1815), crevalle jack *Caranx hippos* (Linnaeus, 1766)) and scombrids (e.g.: little tunny *Euthynnus alletteratus* (Rafinesque, 1810), and Atlantic bonito *Sarda sarda* (Bloch, 1793)). All are reported to prey on round scad or similar small fishes (*Manooch III & Haimovici, 1983*; *Saloman & Naughton, 1984*; *Manooch, Mason & Nelson, 1985*; *Campo et al., 2006*; *Sley et al., 2009*; *Fletcher, Batjakas & Pierce, 2013*).

## Hard bottom habitat

Hard bottoms, or "live bottoms", are rocky habitats and conspicuous geological features on the mainly soft sediment-dominated continental shelf within the SAB. In the Carolina Capes region of the SAB (offshore of North and South Carolina) (*Riggs et al., 1996*) estimates of hard bottom areal coverage vary on both latitudinal and longitudinal bases, with the greatest proportions on the shelf to the south of Cape Lookout, North Carolina (*Parker, Colby & Willis, 1983*; *SAFMC, 1998*). Extensive areas of hard bottom have been mapped and detailed geological descriptions have been published for the arcuate coastal embayments of Onslow Bay (southeastern North Carolina; Cape Lookout to the north and Cape Fear, North Carolina, to the south), and Long Bay (southeastern NC and northeastern South Carolina; Cape Fear to Cape Romaine, South Carolina) (*Milliman, 1972*; *Parker, Colby & Willis, 1983*; *Riggs, Cleary & Snyder, 1995*; *Cleary et al., 1996*; *Riggs et al., 1996*; *Ojeda et al., 2004*; *Denny et al., 2007*; *Taylor et al., 2016*; *Wheaton, 2018*; *NC Division of Marine Fisheries, 2020*). These bays are separated at Cape Fear by the Frying Pan Shoals, an area of shallow, shifting sediments that extend offshore approximately 50 km.

In general, hard bottoms within the Carolina Capes region are emergent areas of sedimentary and biogenic rock (fossiliferous limestones) formed from earlier Pleistocene and Tertiary deposits. They are typically surrounded by much more extensive areas of unconsolidated sediments (mainly sands) deposited during the Holocene. Hard bottoms vary in emergent relief from flat pavements with shallow veneers of sediment (*Riggs et al., 1996*), and typically depauperate epifauna and a near absence of infauna (*Renaud et al., 1996*; *Renaud et al., 1997*; *Renaud, Syster & Ambrose Jr, 1999*), to high-relief scarped hard bottoms (often called ledges) that may be meters above the surrounding sands with vertical or sloped cliffs and combinations of undercuts and overhangs (*Riggs et al., 1996*). Ledges in particular have rich communities of epifauna and rock-boring infauna (*Wenner et al., 1983*).

Hard bottoms provide structural habitat for the settlement of benthic sessile foundation species, including rich assemblages of sponges, scleractinian and octocorallian corals, ascidians, and macroalgae (*Struhsaker, 1969*; *Miller & Richards, 1980*; *Wenner et al., 1983*). The growth of these sessile organisms contributes to hard bottom reef structural complexity. Compared to adjacent soft-sediment dominated areas of the sea floor, hard bottoms create a striking contrast in habitat that supports high richness and abundance of fishes (*Sedberry & Van Dolah, 1984*; *Hopkinson, Jansson & Schubauer-Berigan, 1991*), invertebrates (*Wenner et al., 1983*; *Peckol & Searles, 1984*), and macroalgae (*Schneider & Searles, 1973*; *Schneider & Searles, 1991*; *Schneider, 1976*; *Freshwater & Idol, 2013*). Specifically, hard bottom reefs are home to often large aggregations of small schooling fishes, such as scads (*Decapterus* spp.) and young tomtate (*Haemulon aurolineatum* Cuvier in Cuvier and Valenciennes, 1830), that serve as important prey resources for many pelagic and demersal piscivorous fishes (*Kracker, Kendall & McFall, 2008*; *Auster et al., 2013*).

On hard bottoms off the coast of Georgia, competitive facilitation, an example of a BMII, was observed as schools of round scad and tomtate retreated from multispecies associations of pelagic piscivorous fish towards the seafloor, increasing predation opportunities for demersal piscivorous fishes (*Auster et al., 2009*; *Auster et al., 2013*). Furthermore, *Auster et al. (2009)* observed round scad and tomtate responding to the presence of pelagic predators by reducing nearest neighbor distance and forming more polarized aggregations.

Polarity in fish schools describes the likelihood of alignment and synchronization of movement among individuals (*Shaw, 1978*; *Soria, Freon & Chabanet, 2007*). Increased polarization is a common response to predators for many prey species that form aggregations and has been found to reduce the vulnerability of prey aggregations (reviewed in *Lima & Dill, 1990*). In addition to increasing polarization, prey fishes use several other strategies to reduce vulnerability to predators. Alternative strategies include temporal and spatial changes in forage fish distribution (*Campanella et al., 2019*) and forming aggregations around physical objects, which is hypothesized to be advantageous to prey by serving as a "schooling companion" (*Klima & Wickham, 1971*).

The abundance and taxonomic richness of species on hard bottom reefs and ledges facilitate unique interspecies interactions that are often driven by enhanced feeding opportunities or decreased vulnerability to predators (*McFarland & Kotchian 1982*; *Diamant & Shpigel, 1985*; *Baird, 1993*). For example, associations between blue runner, greater amberjack, and other large piscivorous fishes have been observed during coordinated predatory foraging on hard bottoms (*Auster et al., 2009*). Additionally, facilitative changes in predator behavior and prey reaction are linked to population processes (*Auster et al., 2013*) and patterns of temporal and spatial use of habitat by both predators and prey (*Campanella et al., 2019*). Effects of such associations containing multiple predators on prey mortality has been heavily debated and several studies have reported both additive and reduced predation effects (reviewed in *Sih, Englund & Wooster, 1998*).

## Objectives

In this study, we examined association behavior between sand tiger sharks (STs), round scad (RS), and scad mesopredators (MPs) using video records from a unique, long-term underwater video installation sited on a hard bottom system off the coast of Cape Fear, North Carolina. Based on video observations collected to assess the fish species assemblage (E. Burge, 2020, unpublished data) and video and in situ observations by Burge and others, we repeatedly noticed the unusual association between STs and RS. Consequently, we hypothesized that association behaviors between round scad and sand tigers were more frequent in the presence of pelagic mesopredators than in their absence because the presence of mesopredators represents a potential predation threat for round scad. Strength of the association behavior was believed to be a response to mesopredators that reflects round scad vulnerability. We also suspect that aggregations of round scad may be mutually beneficial to sand tiger sharks by providing camouflage (*Auster et al., 2013*) and increasing predation opportunities on mesopredators. Direct, visual observations of wild behavior of sand tiger sharks are very limited (but see *Smith, Scarr & Scarpaci, 2010*; *Barker, Peddemors & Williamson, 2011*), and this species is currently IUCN Red listed globally as Vulnerable, making research and conservation efforts necessary for protection of this species. Investigating this association behavior may be insightful for a deeper understanding of predation strategies of sand tiger sharks, protective behaviors of a common prey species, and contribute knowledge of trophic dynamics on hard bottom reefs in the South Atlantic Bight.

## MATERIALS & METHODS

### Study site and infrastructure

Video collected in this project are from SharkCam, an underwater, live-streaming camera, publicly-viewable from https://explore.org/livecams/oceans/shark-cam (Explore.org, Los Angeles, CA). The camera is sited beneath Frying Pan Tower (33°29′N, 77°35′W) which is located at the seaward edge of Frying Pan Shoals, approximately 50 km off the coast of Cape Fear, North Carolina. Frying Pan Shoals is within the biogeographic Carolina Province (Cape Hatteras to Cape Canaveral, Florida), a warm-temperate to subtropical zone of the Western Atlantic Region (*Floeter et al., 2008*; *Briggs & Bowen, 2012*; *Briggs & Bowen, 2013*; *Toonen et al., 2016*). The camera is attached to a horizontal support at the base of Frying Pan Tower in about 15 m of water and is surrounded by an expansive area of natural hard bottom (*Riggs et al., 1996*; *NC Division of Marine Fisheries, 2020*) and steel debris, such as large pipes, beams, and gratings, from the exterior of the tower (*Collins Engineers Inc, 2010*) (https://www.youtube.com/playlist?list=PLK1g13VpyT6oYUJL7U3hRPlt2U5L_mcKL). To date, videos from SharkCam have been used to identify 116 temperate and tropical marine fish species (November 2014–February 2020) (*Burge & O'Brien, 2020*).

View Into The Blue® (Boulder, CO) cameras with CleanSweep™ hardware were used for all footage captured (https://www.viewintotheblue.com/). The cameras used (six during the span of this project) featured HD 720p (November 2014–July 2016 footage) or HD 1080p (after October 2016) video resolution, 360° pan–tilt-zoom that rotated on

a pre-determined schedule, or with manual remote control, and a field of view of 62° × 37° (horizontal × vertical). Automatic white balance (color control) was enabled in April 2017 to more closely approximate surface light for color correction. Power is provided by a solar installation atop Frying Pan Tower and data transmission used a line-of-sight radio to shore (*Burge & O'Brien, 2020*).

## Video selection and analysis

SharkCam video files containing sand tiger shark *Carcharias taurus* (STs) records were extracted from a larger video database of all fish species occurrences gathered from non-consecutive, 20-min clips ($n = 1024$) analyzed as part of a larger community analysis project (E. Burge 2020, unpublished data). Video files were recorded during local daylight hours between November 2014 and January 2019.

All 20-min clips containing STs were re-analyzed to confirm shark identification, presence or absence of round scad (RS) *Decapterus punctatus* or a visually indistinguishable species, such as mackerel scad *D. macarellus* (Cuvier in Cuvier and Valenciennes, 1833) or redtail scad *D. tabl* Berry, 1968, and the identification of mesopredators (MPs). Duration of the STs on screen (seconds) was recorded to obtain average observation time.

This more detailed analysis of clips from the larger video database resulted in the rejection of some data records ($n = 26$) because the initial shark identification was not confirmed ($n = 11$ clips), only a fleeting glimpse of STs was available and/or visibility was poor ($n = 12$), or the original video could not be located ($n = 3$). Poor visibility compromised identification of RS or MPs. To these confirmed STs records from the larger database we were able to add 24 clips that contained STs from other video files not previously included in the larger video database by using more recently analyzed clips, personal observations, or reports from citizen scientists recorded with the live video stream on-line (https://explore.org/livecams/oceans/shark-cam). In these added clips, the shark was located and a 20-min clip of video was analyzed as described with the shark centered in the middle. In total, 216 clips with the confirmed presence of STs were analyzed.

Clips with STs were reviewed for the simultaneous co-occurrence of RS and assigned a descriptive category of the association behavior between them using the following categories: no visible interaction (NVI), loosely associated (LA), and tightly associated (TA) (Fig. 1). No visible association was defined as both species moving independently of each other or in opposing directions and with their heads oriented in different directions. Loosely associated aggregations were defined as both species moving in a similar direction, their heads oriented in a similar direction, and RS maintaining an estimated five scad body lengths or more from STs and other school members. Tightly associated aggregations were defined as both species moving as one entity, their heads oriented in the same direction, and RS maintaining less than five estimated body lengths from STs and other school members. These behaviors represent a spectrum of association that we hypothesize correlates with the vulnerability of RS to predation in response to the presence of pelagic mesopredators (MPs); NVI representing the least vulnerable and TA representing the most vulnerable. Video files with multiple sightings of STs were treated as independent occurrences if shark observations were separated by 10-min or greater. Multiple association observations

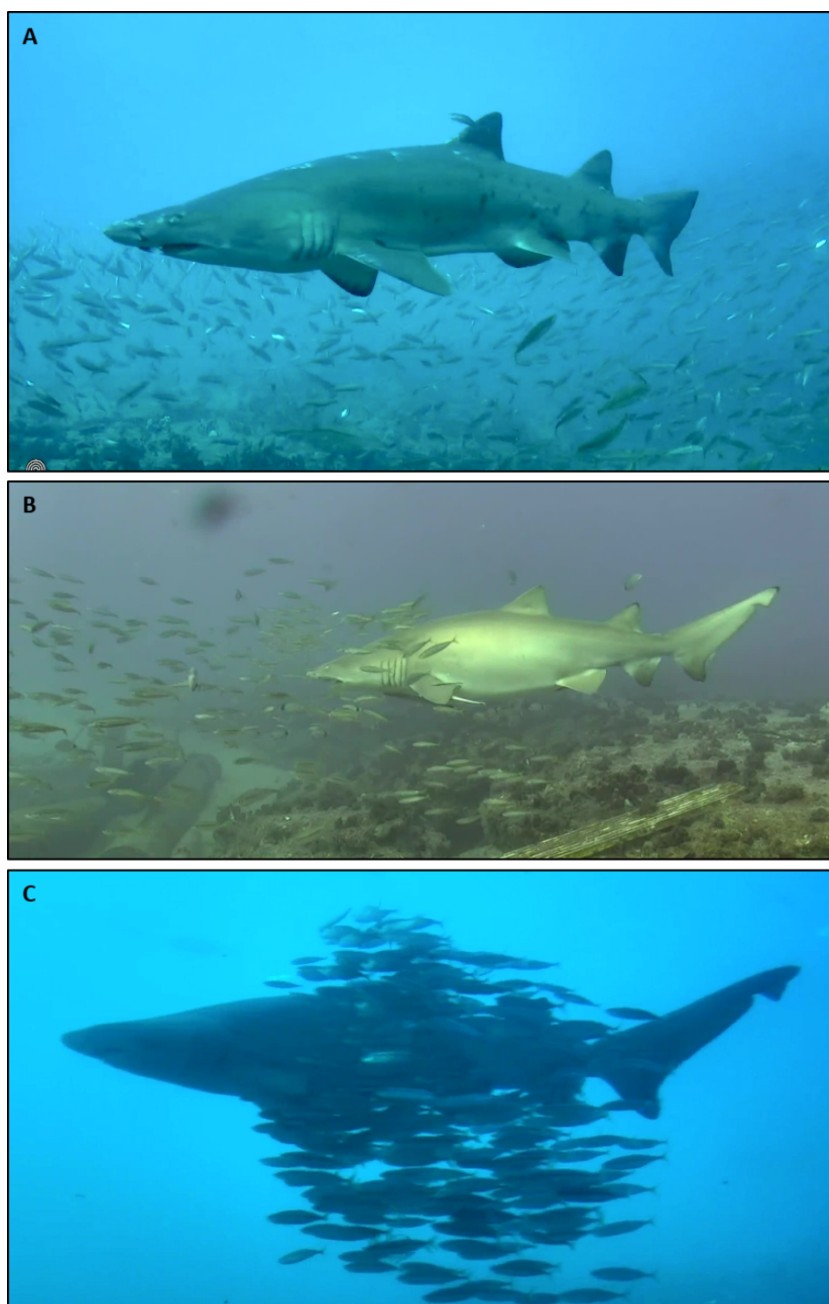

**Figure 1** **Sand tiger sharks and round scad in association behaviors.** (A) No visible interaction (NVI). Image from 18 April 2017, 12:43 EDT, (B) Loosely associated (LA). Image from 11 November 2019, 16:05 EST, (C) Tightly associated (TA). Image from 13 March 2018, 14:14 EDT. Image credits: Erin Burge/Explore.org.

within a 10-min interval were assigned a descriptive category that best described the general association behavior of the recurring individuals. We were not able to individually identify observed STs, however, we have no reason to believe that this association behavior was limited to individual sharks given the multi-year duration over which videos were collected.

Mesopredators (MPs) were hypothesized to be mediators of RS and STs association behavior, and so in order to assess this behaviorally-mediated response, we documented the presence, identity, and occurrence of six pelagic MPs selected a priori within all 20-min video clips that also contained STs. These were almaco jack (*Seriola rivoliana*, AJ), Atlantic bonito (*Sarda sarda*, AB), blue runner (*Caranx crysos*, BR), crevalle jack (*Caranx hippos*, CJ), greater amberjack (*Seriola dumerili*, GA), and little tunny (*Euthynnus alletteratus*, LT). Mesopredators were selected based on preliminary observations of co-occurrence and predation attempts on RS recorded in the larger video database (E. Burge, 2020 unpublished data), published reports that suggest facilitated, or even cooperative, hunting (*Auster et al., 2013*; *Auster et al., 2019*), and shared characteristics of size (individual body size approximately 0.5 m–1.5 m), highly active feeding behavior, and diet overlap (*Manooch III & Haimovici, 1983*; *Saloman & Naughton, 1984*; *Manooch, Mason & Nelson, 1985*; *Campo et al., 2006*; *Sley et al., 2009*; *Fletcher, Batjakas & Pierce, 2013*).

### Data analysis

Hierarchical cluster analysis using Bray-Curtis similarity on presence or absence data was used to illustrate the strength of co-occurrences between STs, RS, and each species of mesopredator in all recorded instances of sharks ($n = 216$). Statistical significance was tested with 1,000 simulation permutations by Simprof ($\alpha = 0.05$) in PRIMER-e 6 (Plymouth Marine Laboratory UK).

Pearson's chi-squared test ($\chi^2$) was used to further investigate whether the frequency of association behaviors between STs and RS were more commonly observed than expected if each category of association had an equal chance of occurring (33%). Equal likelihoods of each association category occurring were assumed to represent a condition in which RS did not benefit from a close physical association with STs. Pearson's $\chi^2$ was also used to test the frequency of association behaviors in the presence of one or more species of mesopredator. The behavior categories were used to represent a continuum of association that reflected the relative vulnerability of RS to pelagic MPs.

Multinomial logistic regression ('multinom', R version 3.5.1) was used to calculate the log-odds of association behavior (i.e., LA and TA behaviors) in the presence or absence of one or more species of mesopredator. This analysis expanded on the results from the $\chi^2$ analysis, which only compared the frequency of association behavior in the presence of MPs, by making predictions of the frequency of association behavior in the presence and absence of MPs. Odds ratios were exponentially transformed to obtain percentage values. We hypothesized that MPs facilitated association behavior between STs and RS, therefore presence or absence data of MPs was used as the independent variable and the association behavior was the dependent variable for this analysis. We expected to see the log-odds of association behavior increase in the presence of mesopredators, therefore the association

**Table 1 Occurrence of sand tiger sharks, round scad, and six mesopredator species in SharkCam videos.** Videos analyzed ($n = 1024$, 20-min clips) were collected November 2014–January 2019. Frequency of occurrence data are for all videos, and those known to contain STs ($n = 216$, 20-min clips). The frequency of STs + RS in all clips was 0.138.

| Common Name | *Species* | Acronym | | Videos containing | Frequency of occurrence | |
|---|---|---|---|---|---|---|
| | | | | | In STs clips ($n = 216$) | In all clips ($n = 1024$) |
| Sand Tiger Shark | *Carcharias taurus* | STs | | 216 | 1.000 | 0.213 |
| Round Scad | *Decapterus punctatus* | RS | | 186 | 0.861 | 0.648 |
| Greater Amberjack | *Seriola dumerili* | **GA** | | 106 | 0.491 | 0.580 |
| Almaco Jack | *Seriola rivoliana* | **AJ** | | 89 | 0.412 | 0.351 |
| Blue Runner | *Caranx crysos* | **BR** | MPs | 61 | 0.282 | 0.210 |
| Crevalle Jack | *Caranx hippos* | **CJ** | | 25 | 0.116 | 0.164 |
| Little Tunny | *Euthynnus alletteratus* | **LT** | | 34 | 0.157 | 0.078 |
| Atlantic Bonito | *Sarda sarda* | **AB** | | 9 | 0.042 | 0.015 |
| | | | | Frequency containing STs + RS + $\geq 1$ MPs = | 0.837 | 0.163 |

**Note.**
MPs are in bold.

category that represented the least degree of round scad vulnerability, NVI, was used as the baseline in the analysis.

## RESULTS

All clips ($n = 216$) that contained one or more sand tigers sharks (STs) were used for hierarchical cluster analysis with Bray-Curtis similarity. Among all clips, 186 (86%) included simultaneous observations of RS and STs and were used in multinomial logistic regression. Of the videos that contained STs and RS, 159 (85%) contained one or more species of MPs and were used in the Pearson's $\chi^2$ (Table 1). Average observation time of STs and RS association behavior within clips was $36 \pm 37.5$ s (mean $\pm$ SD), range 3–254 s, median 26 s.

### Visual observations
In the absence of MPs and STs, RS were commonly observed swimming in unpolarized, distributed schools as they foraged for plankton. Foraging and plankton feeding were inferred from a commonly seen behavior where individuals flex their bodies slightly upward while simultaneously opening the mouth (head-tipping), often with slight lateral adjustments, presumably to ingest individual plankters (https://youtu.be/7_i8hoQXeAU; Table S1).

When STs were not observed and MPs were actively foraging on RS, RS responded by forming denser schools (i.e., reducing nearest neighbor distance), and often retreated to the sea floor or associated with the structure of Frying Pan Tower (https://youtu.be/IesLMb9OStw, https://youtu.be/CTwih5UYaqw, https://youtu.be/CIFLIu2FVfA; Table S1).

During LA behavior with STs, RS exhibited unpolarized schooling behaviors, but maintained proximity and speed with the STs (https://youtu.be/_CIqWVUprmU). During

TA aggregation behavior, RS decreased distance to nearest neighbor and STs, and moved with the STs as a highly coordinated group (https://youtu.be/P37lg7iiDJo; Table S1).

Distinct transitions between behavioral categories were rarely observed, and the few observed transitions captured the movement from LA aggregations to TA aggregations (Fig. 2). During these observations, RS displayed an immediate response (less than one second) to MPs. In some of these occurrences, RS can be described as "pulsating" around the shark, as they transitioned constantly between a LA and TA state (https://youtu.be/9WuEyByf_Pw, https://youtu.be/1Ss-AvAMkVg, Table S1).

## Predation attempts

Regardless of the species involved, apparently successful piscivorous predation seen on SharkCam is exceedingly rare (E. Burge, 2020, unpublished data). During data collection from 1024, 20-min intervals of video (>340 h), we recorded all likely successful fish-on-fish predation events and noted the species involved. Only a small number of observations of STs predation attempts on MPs or other species were made ($n = 5$, Table 2) and these were always on prey species much larger than round scad. Attempted predation was documented on two MPs species, little tunny and blue runner, with two additional attempts on gag *Mycteroperca microlepis* (Goode and Bean, 1879) and red drum *Sciaenops ocellatus* (Linnaeus, 1766). Additionally, STs were only observed making predation attempts on other fishes when tightly associated with RS, with no observations of attempted predation during NVI and LA behaviors. The timing of these events did not suggest a bias towards crepuscular feeding although a very small sample size currently exists (Table 2).

During these rare phenomena, sand tiger sharks made quick lunges toward approaching fishes that appeared oblivious to the presence of the STs, presumably due to shrouds of RS tightly associated with the shark (Fig. 3). Although STs were never observed capturing prey during these attempts, it appeared that schools of RS concealed the presence of STs and attracted scad predators to the STs (see videos in Table 2).

## Descriptive frequencies

Sand tiger sharks are fall-winter-spring visitors seen most frequently during cool months (21.3% overall frequency of occurrence among all months, $n = 1024$; Fig. 4). Round scad are ubiquitous during cool months and appear to depart in May–June and September–October (64.8% overall frequency of occurrence among all months, $n = 1024$; Fig. 4), but are most frequently seen during periods of STs residency. The combinatorial frequency of occurrence for these two species (co-occurrence) was 13.8% within the 1024 videos. Additionally, the frequency of videos (of 1024) that contained STs, RS, and one or more species of up to six different MPs was 16.3% (combinatorial frequency of occurrence for 3+ species) (Table 1).

However, in videos selected because they contained STs (100% frequency of occurrence, $n = 216$), RS were much more likely to be seen (86.1% frequency of occurrence), and the likelihood of this co-occurrence was strongly clustered with the presence of one or more MPs species (Fig. 5) (83.7%, combinatorial frequency of occurrence for 3+ species). Of the six species of MPs selected for this study, four species (almaco jack, blue runner, little

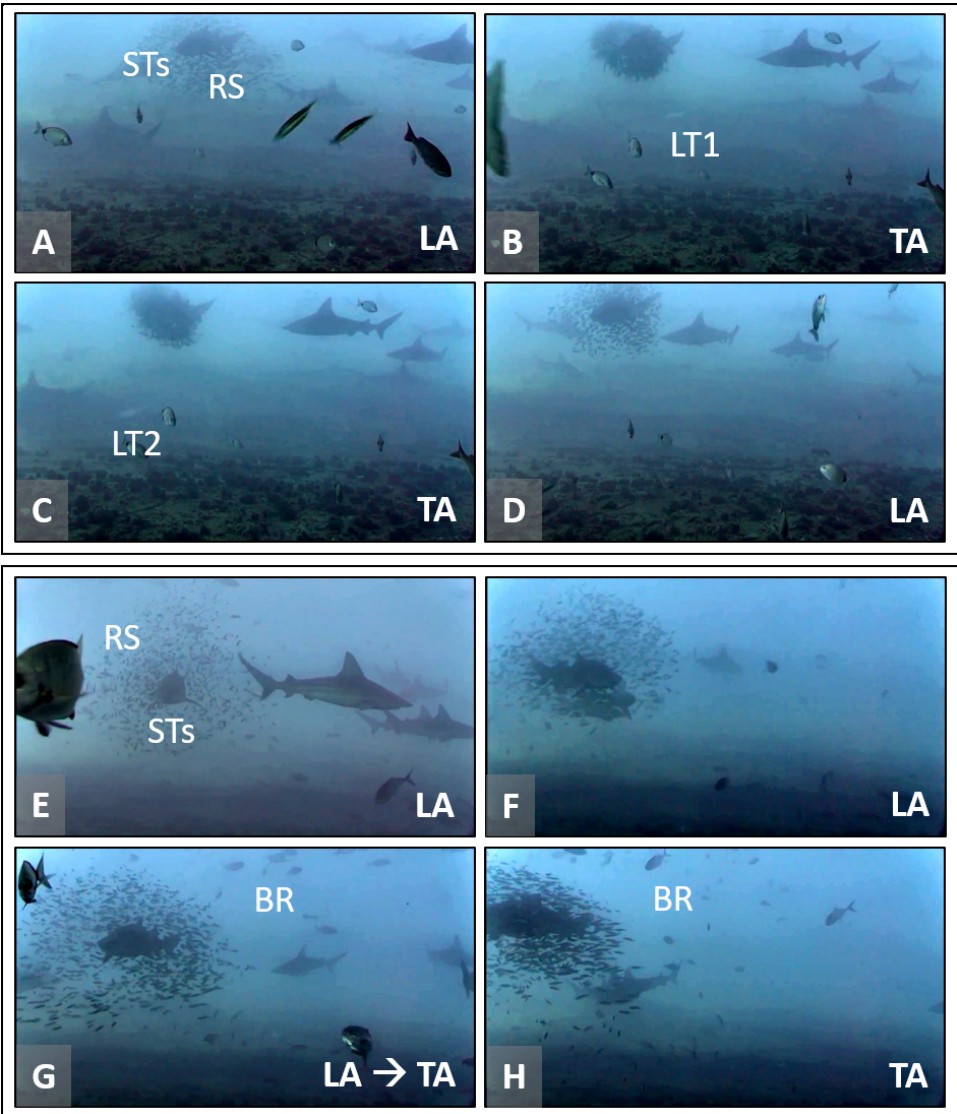

**Figure 2  Association behavior between round scad (RS) and sand tiger sharks (STs) as a response to mesopredators (MPs).** Two sets of screen captures (A–D and E–H, lower left of each image) are representative of transitions between loose association (LA) and tight association (TA) that occur between RS and STs in the presence of MPs, little tunny (LT) and blue runner (BR). (A–D) RS are in LA with a STs. Two LT successively cause a rapid transition to TA. Once attacking mesopredators depart, RS transition to LA rapidly. View video at https://youtu.be/1Ss-AvAMkVg?t=180 (0:03:00). (E–H) RS are in LA with STs. A school of BR approaches and RS rapidly transition to TA behavior. View video at https://youtu.be/1Ss-AvAMkVg?t=925 (0:15:25). Video files are deposited in a public online repository on Zenodo (doi:10.5281/zenodo.4477423). Note that sandbar sharks (*Carcharhinus plumbeus*) are also present in some images. Image credits: Erin Burge/Explore.org.

tunny, and Atlantic bonito) were more frequently observed within the data set of only STs videos ($n = 216$) than the SharkCam occurrence data set ($n = 1024$) (Table 1).
**Table 2   Attempted predation events by sand tiger sharks (STs) on mesopredators (MPs in bold) while in association with round scad (RS + STs).**  All association behaviors were tightly associated (TA). Date (DD Mmm YYYY) and time (U.S. Eastern Standard Time, UTC/GMT -5) are local to the camera. Video time refers to the time within the video clip of the attempt (H:MM:SS). Video files are deposited in a public online repository on Zenodo (doi: https://doi.org/10.5281/zenodo.4477423).

| Prey species | RS+STs | Date of occurrence | Clock time of occurrence | Video time | Video reference |
|---|---|---|---|---|---|
| **Little Tunny *Euthynnus alletteratus*** | TA | 19 Dec 2015 | 1121 EST | 0:00:08 | https://youtu.be/P37lg7iiDJo |
| | TA | 06 Jan 2019 | 1040 EST | 0:00:09 | https://youtu.be/PllHZr-ioeo |
| Red Drum *Sciaenops ocellatus* | TA | 12 Jan 2019 | 1148 EST | 0:19:45 | https://youtu.be/i5wO7ILbbd8 |
| **Blue Runner *Caranx crysos*** | TA | 12 Jan 2019 | 1150 EST | 0:20:52 | https://youtu.be/i5wO7ILbbd8 |
| Gag *Mycteroperca microlepis* | TA | 15 Jan 2019 | 0740 EST | 0:00:07 | https://youtu.be/cfGFAq1cQtI |

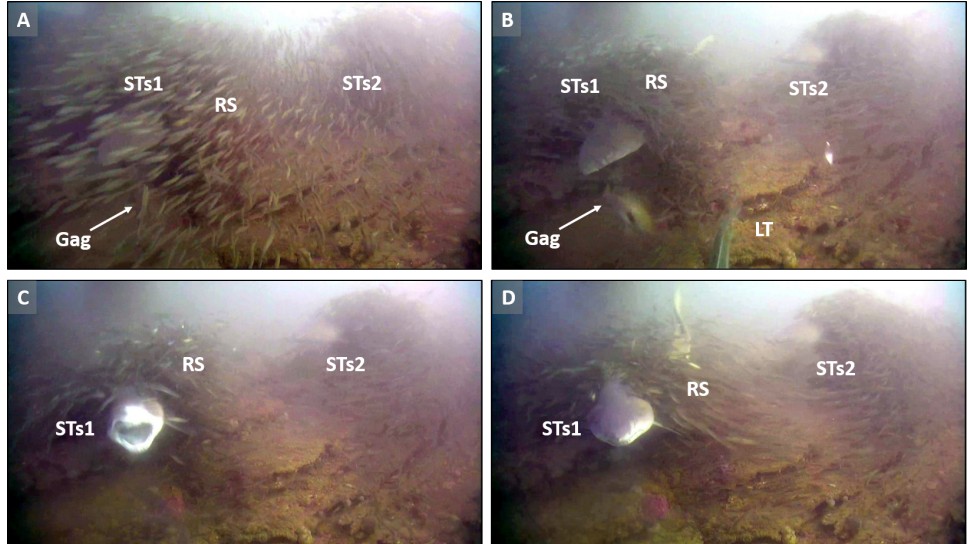

**Figure 3   Association behavior and sand tiger shark predation.** Screen captures (A–D) are representative of tight association (TA) behavior between sand tiger sharks (STs1 and STs2) and round scad (RS). (A) Gag *Mycteroperca microlepis*, approaches, apparently not recognizing the presence of STs1. (B) Simultaneously, a little tunny (LT), approaches shrouded STs1 in what appears to be predation behavior on RS, disrupting the shroud. (C) The STs attempts to prey upon the gag, but (D) is unsuccessful. See the example video for additional information (https://youtu.be/cfGFAq1cQtI). Video files are deposited in a public online repository on Zenodo (doi:10.5281/zenodo.4477423). Image credits: Erin Burge/Explore.org.

## Association behavior

Sand tiger sharks and RS clustered together the strongest (92.5%), while MPs had lower similarities (Fig. 5). Among MPs, GA was significantly clustered with RS and STs and had the highest similarity value of all MPs (63.4%). Almaco jack (AJ; 54.2% similarity) and BR (37.9% similarity) were also significantly clustered with STs, RS, and GA, while LT, CJ, and AB were not significantly clustered within the group.
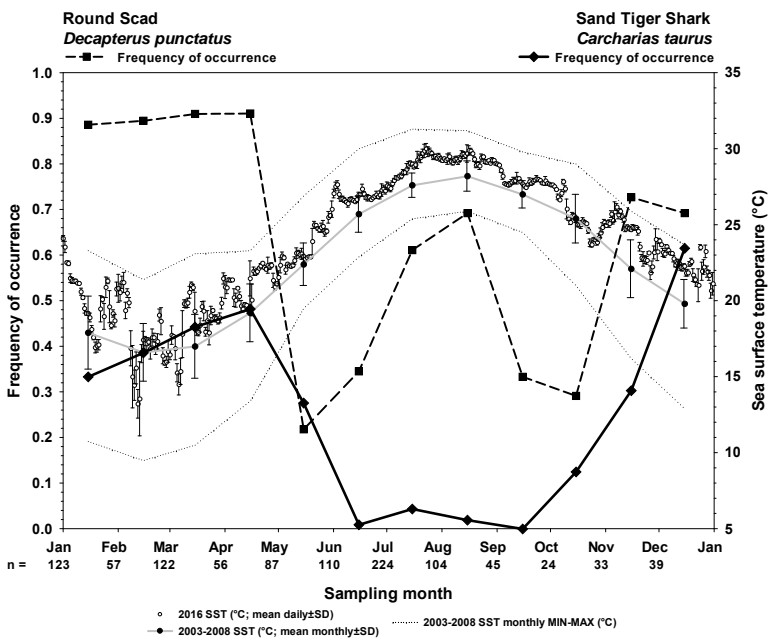

**Figure 4 Frequency of occurrence of sand tiger sharks and round scad indicate the seasonality of their presence at Frying Pan Tower.** Seasonality of sand tiger sharks (STs) and round scad (RS) are represented by frequency of occurrence data (solid line, STs; dashed line, RS) from SharkCam videos ($n = 1024$, 20 min clips) from November 2014–January 2019. Video clips analyzed by month are indicated as ($n =$). Sea surface temperatures (SST) are plotted as 2016 mean daily water temperature (°C) ±SD (open circles), 2003–2008 long term mean monthly SST (black circles on gray line), and 2003–2008 minimum and maximum monthly SST (gray stippled lines) at Frying Pan Tower (data from NOAA NBDC Station 41013 (LLNR 815)–Frying Pan Shoals, NC).

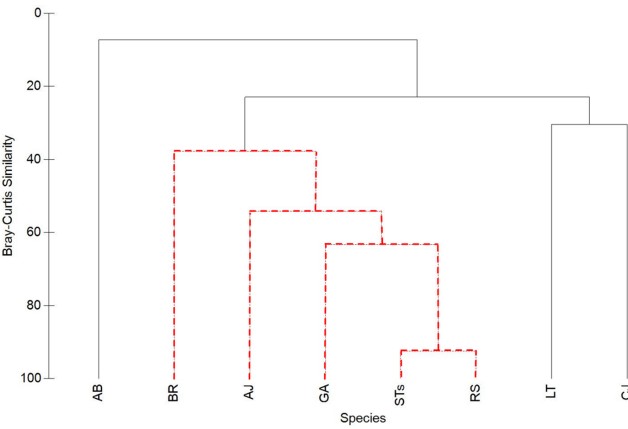

**Figure 5 Cluster analysis of Bray-Curtis similarity to illustrate associations between sand tiger sharks (STs), round scad (RS), and individual mesopredator species (MPs).** See Table 1 for species acronyms. STs and RS cluster strongly (92.5% similarity), while MPs have lower similarities. Clusters containing red-dashed branches are significant (Simprof, $\alpha = 0.05$).

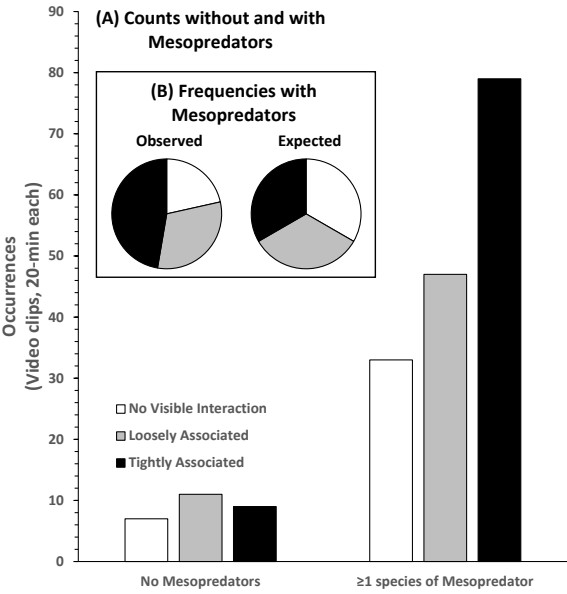

**Figure 6** **Association behavior counts between sand tiger sharks (STs) and round scad (RS).** (A) Without and with the presence of mesopredators (MPs), and observed and expected frequencies (B) of association behaviors in video clips collected from SharkCam ($n = 186$, 20-min clips). Tightly associated behavior between STs and RS occurred significantly more often than expected (Pearson's $\chi^2 = 20.981$ (df 2), $p < 0.000$) in the presence of MPs.

Analysis of the 186 videos containing co-occurrence of STs and RS resulted in 40 observations of NVI, 58 observations of LA behavior, and 88 observations of TA behavior without regard to the presence or absence of MPs (Fig. 6). In the presence of MPs the frequency of LA and TA behaviors increased. NVI was least frequently observed ($n = 33$), and TA behavior was most frequently observed ($n = 79$). There was a significant difference in the observed occurrences of behavioral categories compared to their expected frequencies, assuming equal likelihoods of occurrence in the presence of MPs (Pearson's $\chi^2 = 20.981$; df 2; $p < 0.000$). Compared to no visible association, the log-odds of round scad being loosely associated with STs was 82.9% greater, and tightly associated behavior was 196% greater in the presence of mesopredators, but these results were not statistically significant (LA, $p = 0.727$; TA, $p = 0.218$), likely due to the number of NVI behaviors seen even when MPs were present.

## DISCUSSION

Based on SharkCam underwater video observations conducted over more than 5 years and representing over 340 h of underwater footage, we demonstrate that round scad (RS) are significantly more likely to be associated with sand tiger sharks (STs) in the presence of potential scad mesopredators (MPs), than in their absence. This example of a behaviorally-mediated indirect interaction (BMII) has important implications for trophic energy transfer on hard bottoms in the SAB.

Round scad are one of the most abundant pelagic, planktivorous fish species observed on hard bottom habitats (*Lindquist & Pietrafesa, 1989*; *Rountree, 1990*; *Burge et al., 2012*), and high densities of RS are especially apparent on mid-shelf live bottoms of the SAB in winter (*Hales Jr, 1987*). Under these circumstances, MPs are likely dependent on RS and similar species such as mackerel scad *D. macarellus* and redtail scad *D. tabl* as primary prey resources, especially in cooler months. Given this, it was important to investigate round scad association behavior with sand tiger sharks and be aware that this interaction may relate to trophic transfer and mesopredator control. We have no reason to believe this STs and RS association behavior is unique to the hard bottom habitat off Cape Fear, North Carolina, therefore describing this behavior and its stimulus aids in understanding trophic interactions and prey strategies that influence predation success for pelagic predators.

While the possibility of benefit to STs as increased predation opportunities remains an open question, available observations (see videos in Table 2) provide evidence that STs use this association to perform an undocumented predation strategy. The concealment provided to STs by RS and the attraction of MPs to the shark suggest that this association behavior facilitates the feeding and foraging behavior of sand tiger sharks (Table 2). Feeding behaviors and bite kinematics in captive *C. taurus* have been described (*Ferrara et al., 2011*; *Moyer, Shannon & Irschick, 2019*), but foraging behavior in wild STs remains unclear, although several lines of evidence suggest that crepuscular and night foraging are probable in this species, with a need for substantial additional research (*Hammerschlag et al., 2017*). For example, *Kneebone et al. (2018)* report that juvenile sand tigers are more active at night in a nursery area of Plymouth, Kingston, Duxbury (PKD) Bay, Massachusetts. Using acoustic detections, accelerometer data, geospatial modelling, and field observations *Kneebone et al. (2018)* inferred that foraging behavior may be an important aspect of increased activity.

Admittedly, the limited field of view of SharkCam served as a constraint to fully observe behavior between STs, RS, and MPs because interactions occurred rapidly and in three-dimensional space (see videos in Table S1). In this study, occurrences of MPs outside the perspective of SharkCam undoubtedly influenced events captured on camera, and this is likely responsible for many of the observations of loosely associated (LA) and tightly associated (TA) behaviors that occurred in the absence of MPs (STs+RS-MPs, $n = 27$). One or more species of selected MPs occurred in approximately 84% of videos containing STs and RS (Table 1). Additionally, many other species of potential scad mesopredators are also present at Frying Pan Tower, including a diverse assemblage of demersal piscivores, and other pelagic species not designated as MPs in analyses (Table S2). Recent work by *Brown et al. (2020)*, also on sand tiger sharks in North Carolina, has revealed that the presence of the sharks alters short-term reef fish community richness on shipwrecks, with the prevalence of pelagic mesopredators elevated, and those of demersal mesopredators depressed. They suggest that these differences in richness may be behaviorally-mediated responses attributed to mesopredator optimization of foraging strategy, with sharks as short-term drivers of spatial and temporal community composition for mesopredators. Given this, it is likely that there are few circumstances when aggregations of round scad are not exposed to potential predators. The inclusion of STs+RS-MPs videos in the multinomial logistic regression

likely resulted in the lack of statistical significance ($p = 0.727$ for LA and $p = 0.218$ for TA), despite the relatively large increases in the odds of association behavior as association strength increased (82.9% for LA and 196% for TA). This pattern is supported by the results of the chi-square analysis that show the frequency of occurrence of association behavior increased with association strength ($p < 0.000$, Fig. 6).

In research analyzing predator influence on prey behavior, *Seghers (1974)* hypothesized that guppies formed schools because they were always exposed to predation threats. Constant exposure to predators provides a strong explanation for why round scad are observed forming associations with sand tiger sharks when designated MPs are not seen on SharkCam. While observations of round scad association with sand tiger sharks in the absence of MPs observations may be a result of sampling bias (i.e., MPs were present but not observed on camera), it suggests that aggregation behavior is driven by perceived predation risk assessed by the scad, and further suggests near constant exposure to mesopredators. Experimental manipulation of mesopredators, their prey, and higher trophic-level predator presence have tested perceived predation risk to prey and mesopredators in marine (*Del Mar Palacios, Warren & McCormick, 2016*) and terrestrial (*Gordon et al., 2015*) settings—one commonality that emerges is that the presence of a high trophic-level predator alleviates perceived risk to prey by providing a "refuge effect" associated with behavioral changes in prey.

For example, the mackerel scad *Decapterus macarellus*, an ecologically similar and sympatric relative of round scad *D. punctatus*, associated with goliath grouper *Epinephelus itajara* (Lichtenstein, 1822) presumably to reduce their vulnerability to predation by the horse-eye jack *Caranx latus* Agassiz in Spix and Agassiz, 1831. Mackerel scad were described as forming a dense aggregation around the grouper while under threat by jacks, and the school of scad moved with the grouper as one unit. The authors concluded that this behavior was likely advantageous for mackerel scad by reducing their risk of predation by a mesopredator (jack) that was itself potential prey for the grouper (*Macieira et al., 2010*).

Behavioral descriptions of RS and STs association were created to classify three distinct levels of association based on preliminary observations. Inclusion of a continuous, quantitative variable to measure aggregation strength may have increased the accuracy of identifying round scad responses to mesopredators but was not necessary given how distinct existing behavioral categories were. Nearest neighbor distance is a structural measurement of fish aggregations and it is used to calculate the positional preference of individual fish within an aggregation based on the positions and movements of adjacent fish (*Parrish, Viscido & Grunbaum, 2002*). Evidence supports that synchronization and group coordination are mediated by an individual's interactions with nearest group members (*Soria, Freon & Chabanet, 2007*; *Ballerini et al., 2008*; *Niizato & Gunji, 2011*). We visually estimated nearest neighbor distance of round scad to other aggregation members and to sand tiger sharks to incorporate a measure of aggregation structure into our behavioral descriptions in order to reduce observer bias. Conventionally, model simulations have been effectively used to understand the mechanisms that influence strength, response to stimulus, and coordination of fish schools (*Huth & Wissel, 1992*; *Parrish, Viscido & Grunbaum, 2002*). Observations of RS becoming more aggregated in response to perceived

predation threats are consistent with prior literature (*Rountree & Sedberry, 1991*; *Auster et al., 2009*; *Auster et al., 2013*). Reduction in nearest neighbor distance and increased polarity likely facilitates more efficient communication between school members which enhances group synchrony and coordination (*Rieucau, Fernö & Ioannou, 2015*).

In-situ marine field experiments analyzing fish aggregation dynamics are uncommon, likely given the difficulty to track individual fish in an open setting. Dual frequency identification sonar (DIDSON) has recently become a reliable method to conduct in-situ analysis of fish schooling behavior and has the capability of monitoring the movement of individual fish (*Moursund, Carlson & Peters, 2003*; *Boswell, Wilson & Cowan, 2008*; *Price, Auster & Kracker, 2013*; *Rieucau et al., 2016*). *Auster et al. (2013)* used DIDSON techniques to effectively analyze prey distribution during predator–prey interactions similar to those observed in this study on hard bottoms in the SAB. DIDSON analysis would support more quantitative data on the structure of associations between RS and STs, but was not necessary to understand interactions between RS and MPs and would introduce additional cost for support and maintenance.

Sharks were always the nucleus of association for round scad in our study. We hypothesized that proximity to sharks by round scad, in addition to changes in scad polarization and reduced nearest neighbor distance, served as an additional predation defense for round scad against mesopredators. Although we focused on association as protective for round scad with sharks, it is important to consider other potential explanations for this behavior.

*Fuller & Parsons (2019)* reported observations of association between RS and several species of sharks. In the Gulf of Mexico, aggregations of round scad and another carangid, Atlantic bumper (*Chloroscombrus chrysurus* (Linnaeus, 1766)), associated with blacktip sharks (*Carcharhinus limbatus* (Müller and Henle, 1839), spinner sharks (*Carcharhinus brevipinna* (Müller and Henle, 1839)), and blacknose sharks (*Carcharhinus acronotus* (Poey, 1860)). In situ observations of association between RS and STs had not been previously described prior to the current study, but photographs available online document associations between round scad and sand tiger sharks on North Carolina reefs and wrecks (https://ncaquariums.wildbook.org/gallery.jsp), and these are mentioned by *Fuller & Parsons (2019)*. Their potential explanations for association behavior include protection from mesopredator predation, optmotor responses (*Shaw & Tucker, 1965*), and scatophagy on shark fecal clouds. We did not document the frequency of round scad feeding during associations or observe foraging on fecal clouds, but round scad foraging for plankton was common during observations (Table S1). As fish forage, especially pelagic planktivores like round scad, they become more vulnerable to predation; therefore, future studies should consider foraging benefits and how associations with sharks reduced vulnerability during foraging. It is also important to continue to explore the possible benefit of this association to sand tiger sharks to fully understand how this association influences trophic interactions.

## CONCLUSIONS

Based on underwater video observations from SharkCam, we hypothesized that association behaviors between round scad and sand tiger sharks were more frequent in the presence of pelagic mesopredators than in their absence because the presence of mesopredators represents a potential predation threat for round scad. Scad were shown to be significantly more likely to be tightly associated with sand tiger sharks in the presence of mesopredators, compared to in their absence. This example of a behaviorally-mediated indirect interaction suggests that the presence of a large predator alleviates perceived risk to prey from mesopredators. These results illuminate a previously undescribed behaviorally-mediated indirect interaction with important consequences for trophic transfer and mesopredator control on hard bottom habitats, and supports the usage of long-term underwater camera installations for addressing questions in marine ecology.

## ACKNOWLEDGEMENTS

The authors thank the following institutions and individuals for their time and expertise. Camera installation and ongoing maintenance collaborators included Trevor Mendelow (View Into the Blue, https://www.viewintotheblue.com/; Teens4Oceans, https://teens4oceans.org/), Richard Neal (Frying Pan Tower, https://fptower.org/), Jim Atack (F/V In Sea State), and at Explore.org, Jonathan Silvio, Courtney Johnson, Joe Pfifer, and Candice Rusch (https://explore.org/about-us). Field and tower volunteers included Steve Luff, Matt Davin, Frederick Farzanegan, Sondra Vitols, Cody Sweitzer, David Kish, Saylor Vann, David Wood, Adam Greene, Reed Winn, Doug Noble, Brooke Briza, Dan Madigan, Zach Hart, Brian Atack, and Steven Seeber. Preliminary video observations for the presence of sand tiger sharks were completed by the following participants in the Coastal Carolina University (CCU) QEP project MSCI 399Q Fish Community Monitoring, fall 2015: Chris O'Brien, Randy Fink, John Rainey, and Kyle Gallion; spring 2016: Randy Fink, Rachel Stout, Ashley Sutton, and Kyle Gallion; fall 2016: Lauren Stevens, Olivia Bertelsen, Christine Casterline, Dave Klett, Kelly McConnaughey, and Emily Otstott; spring 2017: Dakota Hughes, Josh Dusci, Kyle Gallion, Megan Brewer, Macy McCall, Tyler McKee, and Derek Bussey; fall 2017: Devon Carey, Jenna Haberle, Theresa Hegarty, Tyler McKee, Gary Sturm, Cody Sweitzer, and Rebecca Wheeler; spring 2018: Tyler McKee, Cody Sweitzer, Layla Elfiki, Megan Wise, and Kyle Gallion; fall 2018: Cody Sweitzer, Faith Saupe, Jessica Pollack, Jared Smith, Amberlynn Fowler, and Kaelen Reed; spring 2019: Catherine Costlow, Peyton Hartenstein, Chloe Keller, Casey Ludwick, Hailey Metzger, Maariyah Najeeb, Max Pagliari, Jessica Pollack, Jessica Sanders, Faith Saupe, and Ryan Ware; fall 2019: Kylie Bostick, Victoria Campbell, Sydney Davis, Shayne Doone, Janina Jones, Casey Majer, Cheyanne Rufener, Jessica Sanders, Faith Saupe, and Jared Smith. Juliana Harding (Marine Science, CCU) assisted with synthesizing temperature data. Lindsey Bell (Mathematics and Statistics, CCU) and Derek Crane (Biology, CCU) provided guidance on probability and statistical analyses. The authors thank Avery Paxton (NOAA National Centers for Coastal Ocean Science), and Rob Condon (Young Scientist Academy,

https://www.youngscientistacademy.org/), and one anonymous reviewer for providing critical feedback and helping to improve this manuscript.

### Funding

Funding for SharkCam was provided by explore, a project of the Annenberg Foundation. Student and faculty support was derived from grants from Coastal Carolina University's Quality Enhancement Plan, Experienced@Coastal program, and a Gupta College of Science Research Fellow award. Additional funding and in-kind donations of time and expertise were provided by View Into the Blue, Teens4Oceans, and Frying Pan Tower. The funders had no role in study design, data collection and analysis, decision to publish, or preparation of the manuscript.

### Grant Disclosures

The following grant information was disclosed by the authors:
Annenberg Foundation.
Coastal Carolina University's Quality Enhancement Plan, Experienced@Coastal program, and a Gupta College of Science Research Fellow award.
View Into the Blue, Teens4Oceans, and Frying Pan Tower.

### Competing Interests

The authors declare there are no competing interests.

### Author Contributions

- Nicholas C. Coleman conceived and designed the experiments, performed the experiments, analyzed the data, authored or reviewed drafts of the paper, and approved the final draft.
- Erin J. Burge conceived and designed the experiments, analyzed the data, prepared figures and/or tables, authored or reviewed drafts of the paper, supported ongoing maintenance at SharkCam, and managed video harvesting and database management, and approved the final draft.

### Data Availability

The raw data are available in Supplementary File 1. They consist of all sand tiger shark, scad, and mesopredator observations (n=216), indexed to unique video clips, with dates and times of observation.

Video clip files referred to in the Results, Table 1, Figs. 2 and 3 are deposited on Zenodo (DOI: 10.5281/zenodo.4477423)

Table S2 lists a description of each video (Descriptions), date (Date of occurrence) and time (Clock time of occurrence) of footage, a timing reference to the description within the video (video time), and a link to Youtube (Video reference) of the same footage.

## Supplemental Information

Supplemental information for this article can be found online at http://dx.doi.org/10.7717/peerj.11164#supplemental-information.

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
