# Peer review of "Association behavior between sand tiger sharks and round scad is driven by mesopredators"

_PeerJ, doi:10.7717/peerj.11164_

## Round 0.1 · original submission · Major Revisions

We have received two reviews for your manuscript. Both reviewers concurred that it was an interesting study. However, they underlined some issues that deserve revisions.

In particular, the introduction needs some work to be better organized and clarified, and both reviewers made useful suggestions on how to do so. Parts of the methods also need some revisions, especially concerning the details of the data used, which should be provided. Some analyses should be clarified, for example the transformation performed on raw data are somewhat unconventional. Finally, the interpretation of the results should also be revised to avoid implying causation.

Reviewer 1 ·

Basic reporting

The authors present a unique and well designed study based on field observations of species interactions with an apex predator. They do a great job placing their work in a broader ecological context, in a subject area ripe for greater attention in ecological research with application in he conservation arena. That said, there are multiple but mostly minor revisions below that would improve this draft. Ultimately, this will make a fine contribution to PeerJ and will be widely cited in the ecology and ocean conservation literature.
Line 29. Suggest “… likely possibility …” is redundant. Perhaps “These interactions potentially yield additional trophic consequences to this unique association and highlight the value …”
Line 34 – paragraph. The authors should include a bit more detail describing hard bottom reefs. These are sandstone and other sedimentary and biogenic materials, generally surrounded by fine-grained sediments, principally sand. These emerge from the rippled sand. Heights above the surrounding seafloor vary, with some a meter or more high, some undercut, some partially buried with sand veneer. Inverts colonize these sites adding to spatial complexity. A few more terms to better describe the geologic and ecologic setting would better inform readers.
Line 48. Not all fishes are pelagic. Dense schools and aggregation of 0-yr tomtate are significant members of schools over reefs, in addition to round and mackerel scad.
Line 51. Suggest specificity and state hard bottom or live bottom “reefs” or ledges.
Line 56. Auster et al. 2009 fine but also see for additional detail: Campanella, F., Auster, P.J., Taylor, J.C. and Muñoz, R.C., 2019. Dynamics of predator-prey habitat use and behavioral interactions over diel periods at sub-tropical reefs. PloS one, 14(2), p.e0211886.
Line 67. See Campanella et al reference above. Changes in forage fish distribution based on predator behavior over 24 hr periods.
Lies 106-113. See ASMFC Coastal Sharks Review for 2017 (http://www.asmfc.org/uploads/file/5d00081cCoastalSharksFMPReview_2018.pdf) and NOAA Fisheries SAFE report (https://www.fisheries.noaa.gov/resource/document/2018-stock-assessment-and-fishery-evaluation-report-atlantic-highly-migratory) for most recent status. Perhaps worth noting that this species is a prohibited species to catch and must be released when encountered as bycatch in fisheries on the US east coast.
Line 114. Are the authors sure these are all round scad? Live bottom reefs at 20-30-ish meters off Georgia often have a mixture (in mixed species schools or segregated within single schools) of both round and mackerel scad (Decapterus macarellus). For example, the Auster et al papers cited within report both species of Decapterus in schools of prey. This issue is not fatal at all for the paper, just a need to acknowledge both species present or possible based on the video observations, especially if IDs were from video only.
Line 161. Usually referred to as “wasp-waist” web or system, not “waisted” - at least in my experience.
Line 184. Campanella et al. reference apropos here too.
Line 201. Suggest “ … using video records from a unique …”
Line 227. Provide finer biogeographic resolution here? This area represents the boundary between the Virginian and Carolinian provinces.
Line 274. Suggest “ … six MPs selected a priori …” or something similar.
Line 291. Why square root transform? Need some brief rationale for this versus no transform at all or any of the multitude of others.
Line 335. Is “entity” the right term? Suggest moving as a “highly coordinated group” or similar?
Line 352. N=? How many are rare?
Line 388. What was the p value? It is useful to report the actual p value to let the reader know the actual level of significance. That is, was p=0.08, or 0.11, or 0.68?
Line 407. Motivation is not what was addressed here, but perhaps “rationale” or similar term(s) to explain association in ecological terms.
Line 474. See Price, V.E., Auster, P.J. and Kracker, L., 2013. Use of high-resolution DIDSON sonar to quantify attributes of predation at ecologically relevant space and time scales. Marine Technology Society Journal, 47(1), pp.33-46.

Experimental design

Well designed. See above.

Validity of the findings

Yes. See above.

Additional comments

Yes. See above.

·

Basic reporting

The authors mostly use clear and unambiguous language in their manuscript. There are several instances, which I outline below where clarification would be helpful. See general comments below.

Experimental design

I recommend that the authors reorganize or rework the introduction to more clearly identify the knowledge gap that seek to fill. The methods require some additional details to ensure reproducibility. See general comments below.

Validity of the findings

See general comments below

Additional comments

Coleman and Burge provide compelling evidence for association behavior between a large coastal shark (sand tiger shark) and baitfish (round scad). Their evidence stems from review of videos from a long term monitoring dataset collected off of Cape Fear, North Carolina. They also demonstrate that the association between the large sharks and small baitfish may be related to mesopredators, including piscivorous fish, such as jacks. The manuscript will make an important contribution to the peer-reviewed literature because it advances current discussions of predator-prey interactions.

Despite the value of the manuscript, I have several concerns that I would recommend the authors address prior to publication. I would recommend that this be conducted through a major revision. I outline my broad feedback below and then provide more detailed feedback.

Broad comment 1: I recommend that the authors rework the organization of the introduction. As written, the introduction provides an incredible amount of detail, but I think that reordering of certain paragraphs so that the first paragraphs are broader and the later paragraphs more study/system/organism specific (e.g., hourglass formation) may facilitate reader understanding and also frame the research more broadly within the field of marine ecology. For example, the current paragraph 3 could become paragraph 1 because it is broadly about multispecies interactions and theoretical ecology. The current paragraph 1 focuses on hard bottom reefs of the Carolinas and seems too detailed and system-specific to occur so early on in the introduction. Theoretical and broad ecological content are mixed into the sections on sand tiger sharks, round scad, and mesopredators. I recommend making these sections parallel to one another (e.g., open with broad content then narrow down) or potentially moving the broad content up in the introduction. There are many ways to accomplish this. To me, it would make sense to open broadly about multispecies interactions, narrow into marine systems, all while setting up the knowledge gap that your study will address. Then, with the knowledge gap identified, you can describe that you require a system and accompanying species to help fill the knowledge gap, which is where the sand tigers, round scad, and mesopredators come in, along with the descriptions of hardbottom reefs (e.g., the system).

Broad comment 2: While there are plenty of details in parts of the methods, other details are lacking or deserve clarification to provide full understanding of the methods. For example, it would be helpful to clarify how videos were selected and retained (e.g., shark presence/absence, mesopredator presence/absence, scad presence/absence). Please also clarify which data were used for which analyses. For example, how were mesopredator abundance values used in the analyses (or were they?)? Was scad abundance incorporated or only presence absence? These details would help me assess the validity of the analyses more thoroughly.

Broad comment 3: Sand tiger sharks are described as apex predators, but I think they should be referred to as large predators instead of apex predators because they likely aren’t filling the role of apex predators (e.g., white sharks, tiger sharks) in this system (see Heupel et al. 2014 in Marine Ecology Progress Series – “Sizing up the ecological role of sharks as predators”).

Broad comment 4: It seems that many of the results are correlational, but in some circumstances are interpreted as causal. Please check language throughout to address this.

Detailed feedback

Abstract

Line 16: Suggest adding an overarching background sentence or two at the start of the abstract to establish the general knowledge gap and broader impact of the study. This would assist readers in understanding the relevance of the study.

Line 16: Explain in the abstract what Frying Pan Tower is. Perhaps you could say “Frying Pan Tower, an artificial tower that rests on natural, hard bottom off Cape Fear, North Carolina…”

Line 25: Perhaps consider a different word for “important” to describe scad. Perhaps, commonly occurring, ubiquitous, abundant, or something similar?

Line 28-29: Could rephrase to be “… round scad association conceals and attracts mesopredators, enhancing predation opportunities” to reduce wordiness.

Line 29: Remove “likely” as it is implied with “possibility.”


Introduction

Line 44 & 48: Reconsider use of word “attract” given that this word is often used in the attraction vs. production debate regarding artificial structures.

Line 56-59: Elaborate on what the effects of the associations are thought to be.

Line 83: Please define “polarized aggregations,” if appropriate.

Line 129: How have the round scad schooling behaviors been described?

Line 130-131: Something seems off about this sentence. The observations don’t really accentuate the affinity but maybe they suggest an affinity. I’d also like to see a citation for the affinity to “physical objects” including the “larger fishes and elasmobranchs,” if possible, or you could qualify the statement to say that the observations suggest an affinity for physical objects, which could (instead of can) include the larger critters.

Line 135-137: This needs clarification. Plenty of studies report on in situ sand tiger shark observations. Do you mean that there are few studies that report on sand tiger and scad associations?

Line 182-184: This sentence seems out of place.

Line 196: Add “stationary” so reads “underwater stationary video.”

Methods

Line 226-227: It seems like a stretch to say that Frying Pan Shoals is so close to the Cape Hatteras biogeographic transition zone. Either drop this sentence or qualify it by providing the distance of the tower from Cape Hatteras.

Line 232: Describe the anthropogenic debris.

Line 245: I am confused about the video clips used for this research versus the larger project database. Please clarify. For example, for the clips used for this MS, were they 20 min clips? Were there 1024 video clips in the overall larger project but then 216 were used for this study? I know this is explained in other parts of the manuscript, such as with the results, but a central place (or maybe a table?) containing the video clip information would be helpful. If a round scad was not present were the clips excluded even if a sand tiger shark was present? What about if a mesopredator was present but not scad and vice versa?

Line 250: Is there any possibility that the round scad could have been misidentified as other types of scad using video footage?

Line 259: Five body lengths of what? Scad body lengths?

Line 267: From what point were the 10-min intervals assigned? For example, were they assigned based on when the shark was first visible in the video clip?

Line 273: “Species composition” doesn’t seem to be the correct term here. Species identity, perhaps? Composition would imply the community composition and thus the abundance and identity of mesopredators? Or, maybe just drop “species composition” as it is implied with the phrase “of six selected MPs.”

Line 290: Presence / absence doesn’t usually get transformed in multivariate statistics, right? If you had abundance, it could be transformed using a square root transformation. Was the mesopredator abundance transformed?

General: What happened if the interaction continued outside of the clip interval?

General: Were videos with sand tigers but without scad excluded, even if they contained mesopredators?

Results

Line 323: I’m unclear on how this statement is supported. I thought videos were excluded if they didn’t include sand tiger sharks but did include round scad?

General: Should the YouTube video links be removed and these videos added as supplementary videos? I’m not sure on journal standards here.

Line 346: What was the number of recorded predation attempts?

Line 349: What does “LT” refer to?

Line 352-356: Wow!! This is fascinating!

Line 358-359: Is this frequency of occurrence for all months or winter months?

Line 385: p = 0??

Line 388-389: Here, you state that the association between sand tigers and round scad was not significantly different in the presence of mesopredators. However, in the first sentence of the discussion, you state that this association is significantly different in the presence of mesopredators? Please clarify this.


Discussion

Line 402: “the ecosystem as a whole depends on round scad…” seems too strong. This study did not examine energy transfer.

Line 404: Similar to comment above, this seems too strong given the correlational evidence. Perhaps “relate to” instead of “effects on.”

Paragraph beginning line 414: This seems redundant with the intro paragraph on underwater video. Perhaps keep the paragraph here in the discussion and remove from intro.

Paragraph beginning line 480: Seems to relate to ideas presented elsewhere in the discussion, so perhaps consider combining with another paragraph.

Line 485-486: Suggest rewording this sentence. I don’t fully understand it, as written.

Line 493-495: Yes! Agreed that this would be very interesting.

General: Sand tiger sharks are assumed to mostly forage at night. Can you please reflect on this? How would this influence the interpretation of your results?

Line 506: Reconsider use of term “apex predator” to describe sand tiger sharks (see broad comment above).

Figures

Figure 2: It would be helpful to know how many videos are from each month. The figure likely isn’t the place for this, but consider including this information in a supplementary table, as it is important for interpreting Figure 2 and statements that sand tiger sharks aren’t observed in some summer months, when I’m wondering if it was actually a detection issue.

Figure 3: Are the data that went into this presence/absence data?

Figure 4: Could this be scaled per clip to be more standardized and aid in comparisons between the left bars (no mesopredator) and right bars (mesopredators)? For example, if we assume (this is my visual estimation…) that the “no mesopredator” bar values are 8, 10, and 9 (from left to right), then the total number of clips without mesopredators is 27. You could then divide the values by 27, so for the third bar it would be 9/27 = 0.33, which would be the scaled occurrence of no mesopredators for tightly associated. You could then repeat this process for all bars. These data could also be displayed in a table of the treatments (NVI, LA, TA) crossed with MP absence and MP presence, if preferred over a figure.


Thank you for the opportunity to review your manuscript. It is a fascinating topic and a compelling dataset. I know your manuscript will be an important contribution to the literature, and I hope that my feedback is helpful.

-Avery Paxton

---

## Round 0.2 · accepted · Accept

I am generally satisfied with the final revisions made on the manuscript.

·

Basic reporting

The authors use clear and unambiguous language and have addressed previous concerns.

Experimental design

The authors have reworked the introduction substantially and added additional details to the methods resolving my previous concerns.

Validity of the findings

No comments to add.

Additional comments

Thank you for the opportunity to re-review this fascinating manuscript. Coleman and Burge have done a commendable job revising their manuscript. They have addressed all of my prior concerns, and I recommend that the manuscript be accepted for publication. This manuscript will make an important contribution to the peer-reviewed literature.

-Avery Paxton